Journal of Data-centric Machine Learning Research (2025)        Submitted 06/24; Revised 11/24; Published 01/25

# Data Acquisition: A New Frontier in Data-centric AI

**Lingjiao Chen**                                                    LINGJIAO@STANFORD.EDU
STANFORD UNIVERSITY

**Bilge Acun**                                                       ACUN@META.COM
FAIR, META

**Newsha Ardalani**                                                  NEW@META.COM
FAIR, META

**Yifan Sun**                                                        YS3600@COLUMBIA.EDU
COLUMBIA UNIVERSITY

**Feiyang Kang**                                                     FYK@VT.EDU
VIRGINIA TECH

**Hanrui Lyu**                                                       HL3616@COLUMBIA.EDU
COLUMBIA UNIVERSITY

**Yongchan Kwon**                                                    YK3012@COLUMBIA.EDU
COLUMBIA UNIVERSITY

**Ruoxi Jia**                                                        RUOXIJIA@VT.EDU
VIRGINIA TECH

**Carole-Jean Wu**                                                   CAROLEJEANWU@META.COM
FAIR, META

**Matei Zaharia**                                                    MATEI@BERKELEY.EDU
UNIVERSITY OF CALIFORNIA, BERKELEY

**James Zou**                                                        JAMESZ@STANFORD.EDU
STANFORD UNIVERSITY

**Reviewed on OpenReview:** HTTPS://OPENREVIEW.NET/FORUM?ID=5ARBFIHXTK

**Editor:** Remi Denton

## Abstract

As Machine Learning (ML) systems continue to grow, the demand for relevant and comprehensive datasets becomes imperative. There is limited study on the challenges of data acquisition due to ad-hoc processes and lack of consistent methodologies. We first present an investigation of current data marketplaces, revealing lack of platforms offering detailed information about datasets, transparent pricing, standardized data formats. With the objective of inciting participation from the data-centric AI community, we then introduce the *DAM challenge*, a benchmark to model the interaction between the data providers and acquirers in a data marketplace. The benchmark was released[1] as a part of Data-Perf (Mazumder et al., 2023). Our evaluation of the submitted strategies underlines the need for effective data acquisition strategies in ML.

---

1. https://www.dataperf.org/training-set-acquisition

# 1 Introduction

Datasets, the cornerstone of modern machine learning (ML) systems, have been increasingly sold and purchased for different ML pipelines (Pei, 2020). Several data marketplaces have emerged to serve different stages of building ML-enhanced data applications. For example, NASDAQ Data Link (Nasdaq, 2021) offers financial datasets cleaned and structured for model training, Amazon AWS data exchange (AWS, 2019) focuses on generic tabular datasets, and Databricks Marketplace (Zaharia et al., 2022) integrates raw datasets and ML pipelines to deliver insights. The data-as-a-service market size was more than 30 billions and is expected to double in the next five years (Daa, 2022).

While the data marketplaces are increasingly expanding, unfortunately, data acquisition for ML remains challenging, partially due to its *ad-hoc* nature: Based on discussions with real-world users, data acquirers often need to negotiate varying contracts with different data providers first, then purchase multiple datasets with different formats, and finally filtering out unnecessary data from the purchased datasets. This is inefficient since negotiation requires tremendous human efforts, while purchasing datasets which are later filtered out leads to a waste of money.

Information opaqueness and lack of principles are the main factors for such an inefficiency. Most data providers are reluctant to offer the full details of their datasets to data acquirers. Consequently, it is challenging for the data acquirers to design principled data acquisition strategies. This is potentially a lose-lose: acquirers fail to identify the desired datasets for their applications, while data providers abandon a large fraction of users and thus lose their revenues. Thus we ask: *how can we design a data marketplace for ML which offers budget-awareness, information and price transparency, and multiple data sources?*

Addressing these important challenges requires not only individual researchers or companies but collaborative efforts from the entire data-centric AI community. To encourage community efforts, we give an in-depth analysis of the existing data marketplaces, and identify four important desiderata of a data marketplace: (i) budget-awareness, (ii) pricing transparency, (ii) useful information sharing, and (iv) multi-provider support. Thus, we design the DAM (data acquisition for machine learning ) challenge, a benchmark for a data marketplace that offers all the desiderata and solicits ML-aware data acquisition strategies. Figure 1 gives an overview of the proposed DAM challenge, which consists of a data acquirer, a broker, and multiple data providers. The data acquirer suggests which model family she is interested in training with the datasets and her own budget. The providers offer their data pricing mechanisms, data summary, and a few samples to the data broker. The broker then decides which (subsets of) datasets to use for the data acquirer and pays the data providers accordingly.

**An example.** Consider a journalist, Alice, who studies the relationship between demographics and economic indicators for an upcoming article. She needs to predict the average annual household income by some demographic features. Datasets with such information exist online, but they are owned by different parties (e.g., companies in different states or cities) and are expensive to purchase. In this scenario, a data marketplace modeled by DAM would allow Alice to be charged only based on her ML task and desired accuracy.

As part of the MLCommons DataPerf initiative (Mazumder et al., 2023), the first launch has attracted promising solutions. Our discussion and analysis of the received strategies

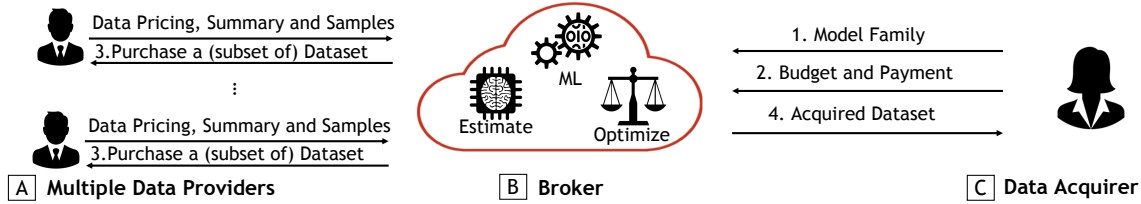

Figure 1: Overview of the data acquisition for machine learning marketplace. It consists of three agents: data providers, a broker, and a data acquirer. The data providers publicly release their pricing mechanisms, data summaries, and a few samples from their datasets. The data acquirer first gives the broker (i) the model family she is interested in training on the purchased data samples, (ii) her own evaluation data, and (iii) the budget she is willing to spend as well as the payment. Next, the broker decides which datasets to purchase as the training data to optimize the model performance on acquirer's data. Finally, it acquires corresponding datasets from the providers and send it back to the acquirer. The DAM benchmark simulates both providers and the acquirer, and ask the participators to construct a broker as good as possible.

underscore the importance of developing data acquisition strategies. In particular, we have found that no single pricing mechanism submitted is universally better than others across all different data market instances considered in our benchmark. Furthermore, it remains challenging to distinguish high quality data for all data market instances. Our contributions are summarized as follows.

> **Main Contributions**
>
> **Overview of existing data markets.** We provide a comprehensive overview of the existing data markets, with a focus on four desiderata including budget-awareness, price transparency, useful information sharing, and multi-provider support.
> **The DAM challenge.** We have designed and implemented DAM, the first data acquisition for machine learning evaluation framework that satisfies all the four desiderata. DAM has been incorporated into the MLCommons DataPerf initiative and attracted many submissions in a few months.
> **Analysis of acquisition strategies received by DAM.** We present an in-depth analysis using the top-ranked acquisition strategies submitted to the DAM challenges.

Overall, we hope this paper lays a foundation for data acquisition in data-centric AI and stimulates a broad range of researchers to tackle important challenges in the area. To encourage more research on this emerging topic, we have released our code at `https://github.com/facebookresearch/Data_Acquisition_for_ML_Benchmark`

## 2 Overview of Existing Data Marketplaces for ML

### 2.1 What type of data acquisition services are there?

Data marketplace for ML is broad and has various forms of commodity that is sold and purchased (see Figure 2). These include labeling services, data acquisition in the model development stage and prediction services in the model deployment stage. Here, the queries are generic and include (i) human labeling services on the dataset, (ii) raw data acquisition, (iii) some data products (such as an ML service) built on top of it. Most data providers adopt (i) and (ii). More recently, more data providers are selling data products (iii) such as ML services. For example, Google uses their own datasets to build vision services, i.e. Google Vision API, which give annotations to user data for a fee (goo, 2023). While all of the mentioned data services are important, we focus on data markets for raw data in this work.

### 2.2 Why is raw data acquisition needed for training ML models?

A natural question is why data acquisition is needed given the abundant amount of publicly available data, such as ImageNet (ima, "2023") consisting of millions of natural images, SQuAD 2.0 containing more than one million English question-answer pairs (Rajpurkar et al., 2018), Common Crawl including petabytes of webpage text data (com, 2023). For many downstream tasks, however, publicly available datasets lack the diversity needed to represent real-world scenarios and frequently suffer from quality issues.

For instance, in the case of Chinese speech recognition, publicly available utterances are mostly recorded in quiet environments, which do not accurately reflect real-world scenarios with diverse noises and delays. Moreover, the speakers in these utterances primarily use standard mandarin, whereas different dialects exhibit distinct pronunciations of the same words or phrases, and some even contain slangs that do not exist in standard mandarin. In the absence of training data that covers these missing contexts, achieving decent performance during inference can be challenging.

Even when publicly available training data covers all possible contexts and domains, the quality of the data remains a concern. Annotation errors are prevalent in many open datasets, such as ImageNet, which can significantly limit the performance of any machine learning models trained on them. In contrast, training on high-quality datasets purchased from professional companies can generate a much higher upper bound on achievable performance.

### 2.3 How does data acquisition for ML happen?

A data marketplace for ML is captured by participants, data or data services, interactions, pricing, and contracts. Participants include data providers, who want to sell their data, data acquirers, who need to acquire data for their own ML applications, and sometimes data brokers, who serve as a middleman between data providers and data acquirers.

For any downstream task, there are often several potential data providers. Data can be sold in bulk as curated datasets or as individual data points. Each provider gives a description of its own dataset, a pricing mechanism, and potentially a few samples from the dataset. There is often some usage term of use associated with the dataset. The most

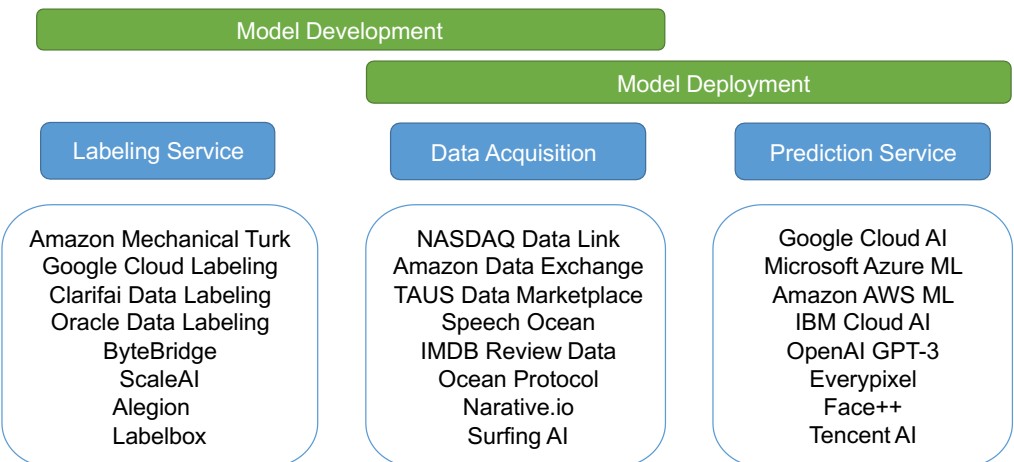

Figure 2: Data Service Types. (i) The long-standing labeling services offer annotations to data (such as images, texts, and audio) provided by the customers. (ii) On the other hand, data acquisition services take the users' description as input, and then returns desired data with or without annotations. (iii) Prediction service emerges as a new data service: it produces machine generations on any given inputs.

important restriction is that the dataset cannot be further sold by the acquirer. This is due to the licensing restrictions.

Table 1 shows a list of some of the existing data marketplaces and categorizes their domain, interaction model, transaction type and pricing model. The main takeaway is that the market is ad-hoc. Different types of data are sold and purchased from different domains. In terms of information shared before purchase, most common practice is to share only metadata information about the dataset or a few data samples. Transaction type also varies: some marketplaces has one time upfront payment, some are subscription based and some charges based on API usage. The prices are sometimes public but in the majority of the markets prices are not advertised publicly and contacting sales is required. Here we expand on the properties in data marketplaces and list the challenges we observe in current marketplaces.

**Roles.** Data provider, data acquirer and broker are the three main roles in the marketplace. A broker is not always necessary – some of the data providers offer their data directly to the acquirers without a third party broker, such as Twitter API (twi, 2023), Nasdaq Data Link (Nasdaq, 2021). On the other hand, brokers can make data access and management easier, especially if tied with a compute platform. For example Amazon AWS Data Exchange (AWS, 2019) and Databricks Marketplace (Zaharia et al., 2022) offer access to a variety of data providers' data to customers through their platform. From acquirers' perspective, having a single platform where multiple data providers data can be found makes it easier to search and find the relevant data. However in the current marketplaces, there are variety of data providers that do not offer their data through a broker platform.

For acquirers this makes access to data harder due to dis-aggregation and for providers it might make it harder to reach to customers.

**Domains.** There are various domains in the data marketplaces such as vision, speech, NLP, finance, healthcare, etc. Some of the marketplaces are not focused on a particular domain; for example AWS Data Exchange (AWS, 2019) and Databricks Marketplace (Zaharia et al., 2022) includes data providers from a broad range of domains. On the other hand some of the brokers are focused on one specific domain, such as Gradient Health (gra, 2023) and Narrative (Ha, 2020). Gradient Health is focused on medical imaging data and gathers patient data from various hospitals. Narrative is focused on demographic and location data gathered from different data providers. Focusing on a particular domain allows these platforms to offer custom features specific for their data type, such as allowing data acquirers to select different attributes from the data and filter it before they make the purchase. For example; Gradient Health allows filtering data by imaging type. Narrative allows filtering people data by age and location. Due to the domain specific nature, each domains would require a different set of attributes that cannot be generalized.

**Interaction Types.** Interaction between the providers and acquirers before making a purchase is critical. The acquirers need information about the dataset properties to validate whether the dataset is useful for their applications. However providers often are not willing to share their dataset prior to purchase and acquirers are not willing to share their use case or models due to confidentiality. This creates the biggest challenge in the marketplace, how to evaluate the value of the data with limited information?

Most of the existing research assumes the providers or acquirers are willing to share their full data (Agarwal et al., 2019) or significant number of data samples (Kang et al., 2023), however in current marketplaces the information shared prior to the purchase of the data is extremely limited. Typical interaction includes data providers to share (i) a few samples from their datasets, (ii) certain meta-data, (iii) summary statistics on the dataset. For example TAUS, Magic Data, Datatang, Core Signal are examples of data provider that share only a few samples from the datasets. AWS Data Exchange, Databricks Marketpace and Speech Ocean provides only some metadata and description of the datasets without any samples.

**Transaction Models.** Popular transaction methods include (i) one-time upfront pricing, (ii) query-based pricing, and (iii) subscription pricing. One-time pricing assigns a fixed price for any given dataset. This works well if the dataset is fixed and relatively small. Query-based pricing allows for sharing a small part of the dataset. For example, one can get 5% of the entire dataset, and only pay for a small amount of dollars. This works when the entire dataset is too large and acquirers cannot afford buying the whole. Subscription pricing gives the users the dataset access only for a fixed period of time.

**Pricing.** This aspect considers whether a data has a fixed price that is visible to all potential data acquirers or has a negotiable price that is not visible publicly. The majority of the marketplaces falls into the second category and they do not show the price publicly. During private price negotiations, providers may offer less price per data sample if acquirer purchases in bulk / more data samples.

**Data Format.** Data can be sold as curated datasets in bulk or as individual data points/samples. Some marketplaces do allow filtering of data based on some features or criteria, however price may not increase linearly with each data point to purchase and buying

| Data Market | Role | Domain Model | Interaction Type | Transaction Model | Pricing Transparency |
|---|---|---|---|---|---|
| AWS Data Exchange (AWS, 2019) | Broker | Varying | Metadata | Upfront/Subscription | Hidden |
| Databricks Marketplace (Zaharia et al., 2022) | Broker | Varying | Metadata | Unknown | Hidden |
| Narrative (Ha, 2020) | Broker | Varying | Metadata | Query based | Hidden |
| TAUS (van der Meer, 2020) | Broker | NLP/Translation | A few samples | Upfront | Fixed |
| PromptBase (pro, 2023) | Broker | Prompts for GenAI | Sample output | Upfront | Fixed |
| Gradient Health (gra, 2023) | Broker | Healthcare | Metadata | Query based | Hidden |
| Snowflake (Snowflake, 2021) | Broker | Varying | Metadata | Query based | Fixed |
| Speech Ocean (spe, 2023) | Provider | Speech, Vision | Metadata | Unknown | Hidden |
| Magic Data (mag, 2023) | Provider | Speech, Vision, NLP | A few samples | Unknown | Hidden |
| Datatang (dat, 2023) | Provider | Speech, Vision, NLP | A few samples | Unknown | Hidden |
| Surfing Tech (sur, 2023) | Provider | Speech, Vision | Unknown | Unknown | Hidden |
| Core Signal (cor, 2023) | Provider | Business, Recruitment | A few samples | PAYG | Hidden |
| NASDAQ Data Link (Nasdaq, 2021) | Provider | Finance | A few samples | Subscription | Fixed |
| Twitter API (twi, 2023) | Provider | Social Media | A few samples | Subscription | Fixed |

Table 1: Examples of data marketplaces and their features. These data marketplaces offer differ in who provides the data, which domain their data comes from, how potential buyers interact with them, the pricing model and transparency.

as bulk can often be more cost effective. Another major challenge in the data marketplaces is the varying data file formats. For a data acquirer, this makes combining data from multiple sources challenging since it requires additional work to convert data formats from different providers into a common format. To address this problem, there are efforts in the industry to unify the data format for ML training, such as Croissant (MLCommons, 2023). Croissant is a *high-level format for machine learning datasets that combines metadata, resource file descriptions, data structure, and default ML semantics into a single file.*

## 2.4 Challenges and opportunities in data marketplaces

Data marketplaces present several challenges, such as ensuring data quality, addressing privacy and security concerns, and creating a fair and transparent pricing system. However, these challenges also present opportunities for innovation. An ideal data marketplace would have several key properties, as shown in Table 2. Firstly, it would have *budget awareness*, where data acquirers can easily understand the cost of the data they are purchasing and make informed decisions about their budget. Secondly, it would have *price transparency*, where data providers can openly communicate their pricing models and data acquirers can compare prices across different providers. Thirdly, it would have *multiple data providers*, offering a diverse range of data sources and allowing data acquirers to choose the best data for their needs. Finally, it would have *useful information sharing*, where data acquirers and data providers can share information and insights to improve the quality and relevance of the data being sold. Yet, none of the existing data marketplaces satisfy all four properties.

In such an ideal marketplace, data acquirers would have access to a wide range of high-quality data from multiple providers, allowing them to make more informed decisions and drive better business outcomes. Data providers, on the other hand, would have a platform to showcase their data and compete on price and quality, leading to increased competition and innovation. Additionally, the marketplace could offer features such as data validation and cleaning, ensuring that the data being sold is accurate and reliable. Overall, an ideal

data marketplace would provide a transparent, competitive, and innovative environment for buying and selling data, ultimately benefiting both data providers and acquirers. With this goal, we designed DAM, Data Acquisition Benchmark for ML (DAM), which we explain next.

| Properties | DAM | AWS Data Exchange(AWS, 2019) | Taus (van der Meer, 2020) | Projector (Kang et al., 2023) |
|---|---|---|---|---|
| Budget Awareness | ✓ | X | X | ✓ |
| Price Transparency | ✓ | X | ✓ | ✓ |
| Useful Info Share | ✓ | X | X | X |
| Multi-Provider | ✓ | ✓ | ✓ | ✓ |

Table 2: Properties of existing mainstream data marketplaces. AWS Data Exchange supports multiple data providers, but their pricing mechanism is often opaque. AWS Data Exchange and Taus gives no budget control. All existing data marketplaces lack a systematic way to share useful information with the potential buyers before transactions. To the best of our knowledge, DAM is the first benchmark for a data marketplace for ML that satisfies all desiderata.

## 3 Data Acquisition for ML Benchmark: DAM

Based on our observations and challenges in the current data marketplaces, we designed a benchmark, Data Acquisition for ML (DAM), with the goal of mitigating a data acquirer's burden by automating and optimizing the data acquisition strategies. In this section, we provide the overall design of DAM along with a concrete instantiation.

### 3.1 Market Setups and Problem Statement

In DAM, we consider a data marketplace consisting of $K$ data providers and one data acquirer. Each provider $i$ holds a labeled dataset to sell, denoted by $D_i$. Note that $\|D_i\|$, the size of these datasets, can vary. To encourage acquirers with varying affordability, data providers allow purchasing subsets of their datasets. For example, one may purchase the entire dataset $D_i$, or only 25% or 50% data points from $D_i$. The price then naturally depends on the number of the purchased samples. Formally, we denote the pricing function for $D_i$ by $p_i : \mathbb{N} \to \mathbb{R}^+$. If $q \in \mathbb{N}$ samples from $D_i$ is purchased, then one needs to pay $p_i(q)$. The pricing function is non-negative and monotone with respect to the number of samples.

**What pre-acquisition information to share with the buyer?** Demonstrations play an essential role in both traditional and data markets. In traditional markets, directly exhibiting the product is a natural way to attract potential buyers. Our discussion with real-world data providers indicates, however, that revealing considerable data instances before the acquirer decides to buy anything is not desired, as the value of the datasets can be lost due to data revealing. Thus, DAM only requires providers to reveal only a small amount (=5) of samples. In addition, summary statistics that describe high-level features of

datasets are often showcased by existing data marketplaces (van der Meer, 2020; Ha, 2020) to attract potential buyers. Thus, DAM also reveals summary statistics on the datasets.

More formally, we use $\mathcal{L}_i$ and $s_i$ to denote the list of shared samples and the summary statistics for the $i$th provider. The data acquirer observes the list of shared samples, the summary statistics and pricing functions, $\{(\mathcal{L}_i, s_i, p_i(\cdot))\}_{i=1}^{K}$. A budget $b \in \mathbb{N}$, a small evaluation dataset $D_b$, and a training model $f(\cdot)$. The distribution of the evaluation dataset is not necessarily the same as the datasets sold by the data providers. In fact, part of the key challenge of data acquisition is to find which data is "similar" enough with the evaluation data before buying it. The acquirer's goal is to identify a purchase strategy $(q_1, q_2, \cdots, q_K) \in \mathbb{N}^K$ and $0 \leq q_i \leq \|D_i\|$ for all $i$, such that the total cost is within the budget $b$, and the accuracy of the ML model $f(\cdot)$ on the evaluation dataset $D_b$ is maximized when it is trained on the purchased datasets. The details of the summary statistics as well as the pricing functions will be given in the next subsection.

## 3.2 Sentiment Analysis on Different Data Providers: A Concrete Instantiation

Here we consider a concrete instance of the above design.

**Setup of the marketplace.** We consider $K = 20$ different data providers. Each of them is selling a dataset for sentiment analysis. Each data point in a dataset $D_i$ is a pair of (i) a feature vector representing the embedding of some text paragraphs, and (ii) a label indicating the nuance of an opinion (e.g., positive or negative) in the text. All providers use the same feature extractors to encode their raw datasets. The quality of data labels also varies across different data providers. The specific data preprocessing details shall be released after the competition is retired. To overcome potential overfitting, we have created five distinct market instances. The structure of each market is identical: 20 data providers, 1 buyer, and the same type of information to share. On the other hand, the data points sold by each provider are sampled from a large-scale data pool using different sampling distributions.The original data pool contains 21 categories. For each data provider, we sample different number of samples from each category. The different samples simulate a diverse marketplace. Each marketplace is also unique due to the varying number of samples from each category.

**Summary statistics**. In our instantiation, the summary statistics contain (i) the 100-quantiles of the marginal distribution of each feature as well as the label and (ii) the correlations between each feature and the label. These summary statistics were selected to offer useful insights on the provider's data while keeping their data secure and private.

**Pricing functions.** Each dataset is worthy $100 and a linear pricing function is adopted. Note that the number of samples within each dataset is not necessarily the same.

**Acquirers' Tasks.** The acquirer holds a small dataset with the same structure (embedding vectors and labels). A logistic regression model is used as the ML model. The acquirer's budget is $150. Each submitter's goal is to figure out the purchase strategy $(q_1, \cdots, q_K)$. After this, each fraction $q_i$ can be converted to the number of samples to purchase via $q_i$ to obtain the number of samples to purchase from each provider.

**Evaluation.** How to quantify the performance of a strategy? For each market instance, we first compute the following score (normalized by 100):

$$\text{score} \triangleq 100 \cdot \left( \alpha \times \text{Accuracy} + (1 - \alpha) \times \frac{\text{budget} - \text{cost}}{\text{budget}} \right)$$

Then we use the average of the five market instances as the final metric. Here, the goal is to maximize the overall accuracy while minimizing the cost. The factor $\alpha$ controls how much budget saving is appreciated. In the existing version of the DAM benchmark, we set $\alpha = 0.98$, encouraging submitters to focus primarily on accuracy.

### 3.3 Solutions

Here, we present the solutions submitted by the benchmark participants. We would like to acknowledge our submitters, including Bilge Acun, Ruoxi Jia, Feiyang Kang, Yongchan Kwon, Hanrui Lyu, and Yifan Sun. Note that the submitted strategies are lightweight, i.e., require a small amount of computational resources (e.g., training some small models to measure similarity), and thus are easily applicable to large-scale datasets.

**Strategy-Single:** The first strategy is to purchase a single provider's data points as many as possible within the budget $b$. To be more specific, this strategy first selects a provider $i \in [K] := \{1, \ldots, K\}$ and purchase $\min(\|D_i\|, n_i)$ data from the $i$-th provider where

$$n_i = \text{argmax}_{x \in \mathbb{N}} p_i(x) \leq b.$$

In DAM, the total price of each provider's dataset is always less than the budget, *i.e.*, $p_i(\|D_i\|) \leq b$, resulting in buying the entire dataset. After the purchase, the remaining budget is exactly one-third of the total budget. We denote this strategy by Strategy-Single-$i$ where $i$ indicates the selected provider's identifier.

**Strategy-All:** The second strategy is to purchase data from every provider with an equal amount of budget for each provider. In contrast to Strategy-Single-$i$, this approach allows us to spend the entire budget, and it is no longer required to select a specific provider. This strategy is expressed as follows. For all $i \in [K]$, we bought $n_i$ data from the provider $i$ where

$$n_i = \text{argmax}_{x \in \mathbb{N}} p_i(x) \leq \frac{b}{K}.$$

We denote this strategy by Strategy-All.

**Strategy-$p\%$** Our third strategy is to purchase data from a subset of data providers by leveraging the distributional similarity between the acquirer and providers. To be more specific, we denote the correlation coefficients between the label and the $k$-th feature within the acquirer dataset by $r_{\text{acquirer}}^{(k)} \in \mathbb{R}$ and set $r_{\text{acquirer}} := (r_{\text{acquirer}}^{(1)}, \ldots, r_{\text{acquirer}}^{(d)})$ where $d$ is the input dimension. Analogously, for $j \in [K]$, a vector $r_{\text{provider},j} \in \mathbb{R}^d$ denotes correlation coefficients between the label and each feature within the $j$-th provider dataset. We then calculate the Euclidean distance between acquirer and provider vectors.

$$Q_j := \left\| r_{\text{acquirer}} - r_{\text{provider},j} \right\|_2^2$$

Figure 4 in the appendix illustrates the distribution of $Q_j$ across the five different markets. It shows there are several providers whose label correlations are more different from those

of the acquirer than others. Based on this observation, we exclude $p\%$ of providers whose $Q_j$ values are larger than others and apply Strategy-All to the remaining providers. We call this strategy Strategy-$p\%$.

**Strategy-RFE (Recursive Feature Elimination)**    Due to the higher dimensional nature of the data (768) and not knowing any of its structure, we reduce the input dimensionality through standard *feature selection with recursive feature elimination (RFE) (Darst et al., 2018)*. Specifically, this backward elimination procedure starts with training the target model with all features. At each time, it removes the feature with the weakest impact on the model's prediction and re-train the model with the remaining features. In our case, the strength of each feature is measured by the corresponding model coefficient's absolute value. This process iterates until the target number of features is reached. The remaining set of features is considered to be most essential to the model's prediction. This helps to find the most important features and refine our analysis to the reduced data.

For each provider's data, the correlation score between each feature to the prediction variable is provided. For ease of elaboration, we refer to this correlation score as *feature relevance* hereafter. Our hypothesis is that if a provider's data is consistent with the acquirer's data and works similarly with target the model, we should observe a high consistency between coefficients of the model trained on the acquirer's data and the feature relevance of the provider's data. For example, for a given feature, if the coefficient of the trained model is positive, which indicates that an increase in this feature increases the chance for the model to predict a positive label, we would expect the correlation between the value of this feature and the label (i.e., feature relevance) to also be positive.

Thus, for features selected by RFE, we first train a logistic regression model on the acquirer's data to obtain the coefficients. A high value on this measure should imply that the data from a provider is more consistent with the validation data such that it better suits the task. Results are visualized in Figure 5 in the appendix. We normalize both coefficients and feature relevance to between 0 and 1 and calculate the *dot product* between the two as the similarity measure. We select the highest valued two datasets, where we select the maximum possible samples for the top 1 dataset and allocate any remaining budget to the second runner-up. Note that this scheme does not take account into the effect of different costs for the data. So we skip the data providers with a higher data cost per sample than the others, which are provider 8 for data markets 2, 3, 4 and provider 9 for data markets 3, 4, 5, respectively.

**Strategy-CoFR (Cosine similarity importance-Feature Relevance)**    As opposed to Strategy-FRE, which examines the top 5 important features selected by RFE, *in this strategy, we calculate the consistency measure (normalized dot product) across all 768 features*. As consistency measure is essentially a proxy to *cosine similarity*, we refer to this strategy as *CoFR (Cosine similarity importance measure-Feature Relevance)*. Results are shown in Figure 6 in Appendix. Same as in Strategy-RFE, we select the top two data providers with the highest correlation to acquirer's data–selecting maximum samples from the first provider, allocating the remaining budget to the second runner-up, and avoiding high-cost data providers.

**Strategy-$L_P$**    Similar to Strategy-CoFR, in this strategy, we calculate the $L_P$ *distance between normalized coefficients of the model trained on acquirer's data and the feature rel-*

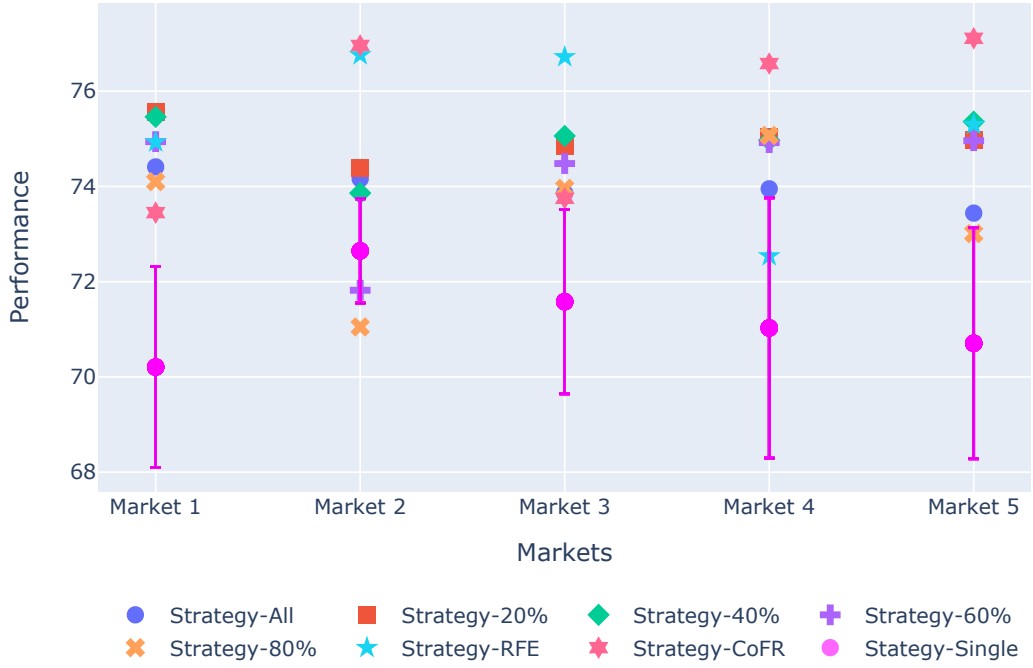

Figure 3: Evaluation of the Proposed Solutions. The pink point denotes the average performance when randomly selecting $i$ and then adopt strategy-single-$i$, while its error bar indicates one quarter of the standard deviation. Strategy-$\ell_p$ is removed for robust visualization, and its performance can be found in Table 3. We observe that no strategy outperforms all others universally. For example, the CoFR approach ranks the first on the second, fourth, and fifth market instance. However, RFE is better for the third market, and Strategy-20% and Strategy-40% rank the top-2 positions for the first market. The variance of Strategy-Single is large. If picking the right single provider, it may achieve the highest performance, which in practice, however, is challenging to do before purchase.

*evance for data from each provider* on all 768 features, where a small distance implies high consistency. We examine $L_2$, $L_1$, and $L_\infty$ distances, respectively. The results are depicted in Figure 7 in Appendix, respectively. Selection scheme is the same as in Strategy-RFE and Strategy-CoFR. Selections for $L_2$ and $L_1$ distances ended up exactly identical.

## 4 Results

We have evaluated all proposed strategies on the five distinct data marketplaces. The results are presented in Figure 3, and the details can be found in Table 3 in the Appendix.

There are several interesting observations. First, there is no universally "best" strategy. For example, the Strategy-CoFR approach gives the best performance on the second, fourth and fifth data marketplace. However, Strategy-RFE is better for the third market, and Strategy-20% and Strategy-40% are the top-2 for the first marketplace. This underscores the importance of carefully customizing the data acquisition strategies for different marketplaces. Second, there is a large variance for the Strategy-Single approach. In fact, we observe that Strategy-Single-20 is often the best strategy, and Strategy-Single-3 and Strategy-Single-8 lead to limited performance. Detailed list of results are shown in Table 3 in Appendix. In practice, however, it is challenging to predict which data provider leads to the best or worst performance, and randomly picking one ends up with limited performance.

## 5 Looking Forward

### 5.1 Alternative Data Acquisition Benchmark Designs

There can be various alternative benchmark designs that are useful for data marketplaces. In this section, we discuss some of the useful scenarios we identified.

**Pre-acquisition Evaluation:** Given the limited information provided by data providers, an important challenge faced by an acquirer is to estimate how well a model trained on the provider's data performs on the acquirer's data seen during deployment (in terms of accuracy, f1, mAP, etc.). This benchmark would enable the acquirer to get an estimate of the value of the data with a more direct metric.

**Iterative Data Acquisition:** This work lays the design foundation of data acquisition benchmark by focusing on the one-shot acquisition strategies, i.e., first observing all available information in the data market and then determining what to purchase once. This captures several real-world applications, but many ML use cases are iterative in nature. Hence, the data acquisition process involves multiple iterations, too. For example, to train a health care assistant, one might first purchase a few thousand anonymous electronic health record data, and then realize the shortage of data on Asian patients. After gaining these new data and retraining the model, she/he may notice the need for elderly or female data. *Iterative data acquisition* raises many interesting questions: how to allocate the budget among different iterations/rounds? How to leverage purchased datasets to help decide which new datasets to buy? And how to balance exploration and exploitation in an iterative acquisition process?

**Data Labeling Selection:** Data labeling is important as many machine learning techniques are supervised. An alternative benchmark could focus on data labeling where data providers sell data labeling services instead of the raw datasets. This challenge is tailored for dataset acquirers to answer the question: given a fixed budget, how should an acquirer decide which data providers to query for the data labels, and how many labels to query from their unlabeled datasets?

**Mechanism Design for Data Transactions:** So far all challenges are tailoring to dataset acquirers. What does a data acquisition challenge look like from the perspective of data providers? Perhaps the most important question for providers is *how to design an effective mechanism to sell their datasets. How to enable quantitative measures to enable acquirers the tool to evaluate how useful a dataset is.* At the same time, the evaluation mechanism must also ensure that the acquirer cannot infer individual data points in the provider's dataset. How should the price of a dataset be determined to maximize revenue?

**Dynamic Data Acquisition:** This work presents the data acquisition benchmark design by assuming a fixed value function for data samples in static datasets on the marketplace. While the static dataset assumption represents certain real-world use cases, in many ways, machine learning datasets are dynamic in nature. For example, real-time data is constantly curated to capture evolving user interests or current events in modern deep learning recommender algorithms (Zhao et al., 2022). Another example is federated learning, where data samples are continuously generated by a large pool of distributed client devices. Interesting opportunity arise for such dynamic, highly distributed machine learning environment — *what should a marketplace look like for data aggregation through federated learning? How should data value be specified to incentivize data sharing through federated learning participation?*

**Privacy and ethics issues.** We acknowledge that this paper focuses on establishing the acquisition foundations and analyzing predictive accuracy achieved by acquisition strategies, and thus data privacy and ethics are out of the scope of this paper. Protecting data privacy and ethics raises several open research questions, such as (i) how to anonymize identities while keep the data quality, (ii) how to update (e.g., delete) a user's record in the purchased datasets upon request, and (iii) how to avoid biases towards sensitive features (e.g., gender or race) in acquisition strategies. We refer the interested readers to (Vincent and Hecht, 2021) for a more detailed discussion on this topic.

## 5.2 A Common Data Format

The data acquisition benchmark design is our first step towards a consistent evaluation mechanism to assess and differentiate value of data. However, working with machine learning datasets on existing marketplaces is needlessly hard because each dataset comes with its unique file organization. The data format fragmentation across datasets on the marketplace and the lack of metadata tailoring to the datasets is a practical challenge faced by realistic data acquisition solutions.

To enable effective data acquisition at-scale, we need standard data formats for machine learning. When data formats and metadata are standardized across datasets in a marketplace, evaluating the value add-on of new datasets is easier for data acquirers. It will also accelerate the development of data acquisition algorithms – a key contribution of this work. Finally, it improves data quality and reduces the ever-increasing storage cost for AI data. We believe a common data format is key to propel the field and an enabling factor to effective data acquisition decisions.

## 5.3 Open-access datasets and data markets

Open dataset repositories are complementary to the data marketplaces in three ways. First, they can be viewed as selfless data sellers in the market, i.e., these who ask for no payment to access their datasets. Second, datasets offered by data sellers can be viewed as a cleaned version of the open datasets. For example, many images' labels in ImageNet were shown to be wrong (Northcutt et al., 2021), and a data seller may sell a corrected version of ImageNet for profit. Thirdly, open datasets can serve as a performance verification tool for the data acquirer.

## 6 Related Work

**Active Learning:** Active learning deals with the problem of iteratively selecting data points from a large (usually unlabeled) data pool (to be labeled) (Settles, 2009; Zheng and Padmanabhan, 2002). It is based the setup that the ML model developer do have access to the full unlabeled data pool. This problem setup is not applicable to data acquisition from real life data marketplaces as the full data is not visible to the acquirer.

**Data Acquisition:** Existing research on data acquisition is not reflective of the real data markets. For example one study proposes a data purchase algorithm for ML model training where the data is labeled and the price per data instance is fixed (Li et al., 2021). The work relies on iterative data sampling and purchase however as we discussed earlier, in some datamarkets datasets are sold as bulk instead of individual samples.

Other work suggested *Try Before You Buy* approach provides an efficient algorithm for evaluating a list of datasets for ML and then deciding which one to buy (Andres and Laoutaris, 2022). However it relies on full access to the datasets, which is not reflective of the real data markets.

An alternative way to solve the problem of limited information share between providers and acquirers was proposed through a platform that incentives the providers to share their data in exchange for rewards (Fernandez et al., 2020). Whether such a platform can be effective or not in real markets is not clear.

**Data Pricing for ML:** There is an increasingly growing interest in analyzing and designing data pricing mechanisms for ML (Chen et al., 2019; Liu et al., 2021; Chen et al., 2022; Agarwal et al., 2019; Pei, 2020; Cong et al., 2022). For example, (Agarwal et al., 2019) designs a data marketplace for exchanging ML training data with a focus on fairness. (Chen et al., 2019) proposes a model-based pricing mechanism which offers arbitrage-freeness and revenue optimality. Furthermore, (Liu et al., 2021) integrates this mechanism with differential privacy. We refer interested readers to comprehensive surveys on this topic (Pei, 2020; Cong et al., 2022). Data pricing mechanism designs often aim at optimizing utility of data sellers, while our benchmark focuses on aiding the data acquirers in the existing marketplaces.

**Data Valuation:** Data valuation studies the contribution of individual data points to the trained ML models (Jiang et al., 2023). Among others, Shapley value (Ghorbani and Zou, 2019) has become the de facto approach to quantify data values. Several techniques have been developed to make it more computationally efficient on specific learning models (Jia et al., 2019b,a; Kwon et al., 2021), extend it to take statistical aspects of data into account (Ghorbani et al., 2020), and twist it for noise reduction (Kwon and Zou, 2021).

Our work is orthogonal to earlier work like Shapley values in three ways. First, existing data valuation techniques such as Shapley values require white-box access to all training instances, while our framework only requires meta information such as summary statistics. Second, Shapley values force the sellers to share an acquirer-defined revenue, while our framework allows sellers to determine their own pricing functions. Third, computing Shapley values is often prohibitively high if not impossible especially for large-scale datasets, while the acquisition strategies studied in this paper are computationally efficient and thus applicable to large-scale data marketplaces.

## 7  Acknowledgements

Thank you to Ce Zhang, Mostafa Elhoushi, Luis Oala, Max Huang, Sudnya Diamos, Danilo Brajovic, Hugh Leather and the DataPerf organizers who gave feedback in designing the data acquisition challenge and during alpha testing. Thanks to Rafael Mosquera for his feedback on the benchmark and his efforts in supporting DAM in DynaBench.

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
