# OpenReview forum: "Data Acquisition: A New Frontier in Data-centric AI"
_DMLR — Accepted by DMLR_

### Review · Reviewer_TxVh · 2024-07-30

**Recommendation:** 3
**Confidence:** 2

**Summary Of Contributions:**

The paper introduces a benchmark called DAM for data acquisition in machine learning (ML). It looks at problems in current data marketplaces and suggests ways to buy data more effectively.  The paper aims to stimulate research in data-centric AI and improve the efficiency of data transactions for ML applications.

**Strengths:**

See above.

**Audience:**

Yes

**Broader Impact Concerns:**

I think the paper doesn't adequately address potential ethical concerns (there are some mentions of it, but I do not see a concrete solution) related to data marketplaces, such as privacy issues, data ownership, and the potential for creating or exacerbating biases in ML models through selective data acquisition.

**Claims And Evidence:**

Yes, to some extent. It would have been better to have diverse pricing models and use cases.

**Datasets And Benchmarks:**

I did not see a link to the DAM challenge (benchmark) webpage. I suggest putting it as a footnote with the abstract.

**Extended Submissions:**

N/A

**Limitations:**

The focus on a single use case (sentiment analysis) and simplified pricing model may limit its generalizability to more complex ML scenarios. The benchmark doesn't address the challenges of acquiring data for pretraining large models or handling multi-task learning scenarios.

**Requested Changes:**

Include a discussion on open dataset repositories (e.g., Zenodo, Huggingface, PhysioNet, Kaggle) and their role in complementing marketplace structures.

Address the role of pubic datasets from government and privately funded research in data acquisition and marketplaces.

Discuss the applicability of strategies (e.g., Strategy-CoFR and Strategy-LP) to large-scale datasets and model pretraining scenarios.

Expand on pricing strategies for more complex scenarios, such as training foundation models that can serve multiple downstream tasks and performance of those tasks can vary.

**Strengths And Weaknesses:**

Strengths: The paper addresses an important and long-standing issue in ML, providing a good overview of data marketplace challenges. The proposed DAM benchmark offers a structured approach to evaluate data acquisition strategies.
Weaknesses: The pricing model and use case (sentiment analysis) are simplistic, potentially limiting the benchmark's applicability to real-world scenarios. The paper lacks discussion on open data repositories  and government-funded publicly available research datasets.

---

### Review · Reviewer_nZRh · 2024-07-30

**Recommendation:** 4
**Confidence:** 3

**Summary Of Contributions:**

The paper presents an exploration of the challenges and opportunities in data acquisition for machine learning systems, focusing specifically on data marketplaces. The authors make the following key contributions:

- Analysis of Existing Data Marketplaces: The paper provides an overview of existing data marketplaces and highlights their features and limitations such as lack of transparency, unstandardized data formats, and complex pricing models.
- Evaluation of DAM Benchmark submissions: The authors evaluate data acquisition strategies submitted to the Data Acquisition Benchmark (DAM).
- Proposed Improvements: The authors propose desirable features for an alternative Data Acqusition Benchmark and in extension desired properties of future data marketplaces, addressing the shortcomings mentioned in the initial analysis.

**Strengths:**

s. S1-S3

**Audience:**

Yes

**Broader Impact Concerns:**

No concerns.

**Claims And Evidence:**

The claims made in the submission are generally well-supported by the evidence provided.

**Datasets And Benchmarks:**

No concerns.

**Extended Submissions:**

This submission appears to be an original work, not an extended version of a previously published paper.

**Limitations:**

- L1 Benchmark Improvements: The paper does not offer concrete / actionable plans for improving the DAM benchmark, however, given the extent of the paper this  be considered out of scope.

**Requested Changes:**

Have all been addressed

**Strengths And Weaknesses:**

- S1 The paper provides a thorough analysis of existing data marketplaces, identifying key challenges such as limited transparecny, inconsistent data formats, heterogeneity of pricing and purchasing models, as well as domain-specificness of data.
- S2 Evaluation of submission results to the DAM benchmark offers a glimpse into the effectiveness of different purchasing strategies that a data acquirer could leverage in a standardize data market place scenario
- S3 FThe discussion on future benchmark designs and the need for standardized data formats provides valuable insights for ongoing research in data-centric AI.

- W1 Some doubt on how well it represents the real world.
- W2 I think the paper would benefit from a more concrete implementation plan for those improvements in the DAM benchmark, but I think it is fair to consider that out of scope for this particular paper.
- W3 Ethical and Privacy Issues: The paper could delve deeper into the ethical and privacy implications of data acquisition, especially given the sensitivity of data in some domains.

---

### Review · Reviewer_ep1p · 2024-11-03

**Recommendation:** 3
**Confidence:** 2

**Summary Of Contributions:**

This paper presents a study on the challenges and opportunities in data acquisition for ML systems. The article highlights the lack of consistent methodologies and platforms offering detailed information about datasets, transparent pricing, and standardized data formats.

In response, this paper introduces the DAM benchmark, a model designed to optimize the interaction between data providers and acquirers. The analysis of the submitted strategies for the DataPerf benchmark underlines the need for effective data acuisition.

Alternative benchmark designs and open problems are further discussed.

I thought the topic of this article is interesting and so I started with excitement to read the article. However, I am afraid the current version fell short of the vision that the authors are trying to convey. In particular, I find the quality of the article to be pretty low due to numerous typos and citation issues. The introduction as well lacks sufficient details for me to tell what exactly is the contribution of this writing.

**Strengths:**

This paper provides a thoughtful survey on the current state in data acquisition. However I'm not completely sold on the prospect of data acquisition being a new frontier in data-centric AI (I find it a lack of evidence to justify this at the moment).

**Audience:**

Yes

**Claims And Evidence:**

I like the question that the authors posed earlier "how can we design a data marketplace for ML which offers budget-awareness, information and price transparency, and multiple data sources?" However, it wasn't clear to me that this paper "this paper lays a foundation for data acquisition in data-centric AI" as the authors have hoped.

**Datasets And Benchmarks:**

The authors introduce a data acquisition benchmark however I did not sufficient details on how this is collected or how the benchmark is organized and will be maintained.

I'm on the fence in terms of leaning to accept and leaning to reject regarding this paper. I think the main idea is nice and it would be useful to the community, however the current writing fell short of the vision that the author is imagining in my honest opinion. I think a significant revision is required before the paper is ready for publication, including fixing all the issues with writing/references, adding more examples to better explain the concepts, strategies, etc., and providing details on the benchmark, including open-source code, data collection, maintenance.

**Extended Submissions:**

N/A

**Limitations:**

How does this new framework relate to the earlier work on data valuation like Shapley values? Would it be possible to spell out the connection more concretely?

**Requested Changes:**

- "et al." in reference list
- Typos: page 3, "we focus data markets for raw data in this work."
- Citation style: missing a bracket, e.g., "Datasets, the cornerstone of modern machine learning (ML) systems, have been increasingly sold and purchased for different ML pipelines Pei (2020)." -> Datasets, the cornerstone of modern machine learning (ML) systems, have been increasingly sold and purchased for different ML pipelines (Pei (2020)).
- Introduction lacks information about relevant articles on similar topics.
    - It would be useful for have a summary of the results in the introduction and in the abstract. By reading the introduction alone I do not have sufficient information to tell what is the main contribution of this article.
- On page 7, table 7 went very much over length. Similarly, for table 1.
- Can you give more examples of what you mean by acquiring data from providers/brokers? For instance in figure 1, can you provide a real life scenario to explain this concept?

**Strengths And Weaknesses:**

S1) The topic is very relevant and this article can potentially generate discussions about the prospect of data aquisition.

S2) An overview of current practice, including different roles of participants, providers, acquirers, are provided. Links to existing data markets are also provided, including their pricing strategy.

W1) I find that the title sounds a bit like an oversell of the actual contribution from this writing. The quality of writing is relatively low because there are numerous typos, incorrect citation styles, and incomplete reference list information.

W2) It is mentioned that a benchmark is introduced but I did not see the link to the data or to the code in the paper. Please correct me if I missed it.